# Non-Induction Basiliximab to Facilitate Renal Recovery via Temporary Tacrolimus Cessation in Cardiothoracic Transplant Patients

**DOI:** 10.3390/medicines12030022

**Published:** 2025-08-28

**Authors:** Tanner A. Melton, Molly W. Fenske, Stacy A. Bernard, Kristin C. Cole, Kelly M. Pennington, Adley I. Lemke

**Affiliations:** 1Department of Pharmacy, Mayo Clinic Hospital, Rochester, MN 55905, USA; fenske.molly@mayo.edu (M.W.F.); bernard.stacy@mayo.edu (S.A.B.); lemke.adley@mayo.edu (A.I.L.); 2Department of Quantitative Health Sciences, Mayo Clinic Hospital, Rochester, MN 55905, USA; cole.kristin@mayo.edu; 3Division of Pulmonary and Critical Care Medicine, Mayo Clinic Hospital, Rochester, MN 55905, USA; pennington.kelly@mayo.edu

**Keywords:** basiliximab, heart transplantation, lung transplantation, acute kidney injury, infections, graft rejection

## Abstract

**Introduction:** Reversible and irreversible nephrotoxicity are known complications of tacrolimus. Approaches to reduce the incidence of nephrotoxicity include the reduction or avoidance of tacrolimus but must be weighed against risk of rejection. Infrequently, basiliximab has been used outside of the induction period to facilitate temporary tacrolimus cessation in the setting of acute kidney injury (AKI). **Objective:** The primary objective of this study was to describe renal recovery after temporary tacrolimus cessation with non-induction basiliximab (NIB) compared to a matched cohort. **Methods:** We conducted a single-center study of adult cardiothoracic transplant recipients that received basiliximab beyond post-operative day 7 for temporary tacrolimus cessation in the setting of AKI between January 2019 and November 2023 and matched them to acontrol cohort. **Results:** Twelve patients underwent temporary tacrolimus cessation with NIB. In total, 7 (58%) patients achieved initial renal recovery at tacrolimus resumption compared to 15 (42%) patients in the matched cohort at an equivalent time point. No difference between treated rejection (17% vs. 19%, *p* = 0.80) or infection (75% vs. 50%, *p* = 0.32) was observed between tacrolimus cessation and its matched cohort. **Conclusions:** The use of NIB for tacrolimus cessation can allow for potential renal recovery after an AKI or in patients at risk of AKI. This approach does not appear to significantly increase the risk of rejection but may increase the risk of infection in the long term.

## 1. Introduction

The calcineurin inhibitor (CNI) tacrolimus is used for maintenance immunosuppression in >92% of heart and >80% of lung transplant recipients [1,2]. Immune suppression occurs through the inhibition of calcineurin, leading to impaired transcription of interleukin 2 (IL-2) and inflammatory cytokines responsible for the activation and proliferation of T cells [3]. Known complications of tacrolimus are reversible acute nephrotoxicity and irreversible chronic nephrotoxicity. Reversible acute nephrotoxicity caused by afferent arteriole vasoconstriction can occur at any level of exposure [4]. Chronic vasoconstriction, direct tubulointerstitial injury, and glomerulosclerosis are attributed with the development of irreversible nephrotoxicity [3]. Currently, the reduction or avoidance of tacrolimus is an approach to decreasing the incidence of chronic nephrotoxicity. This may be accomplished through reduced tacrolimus trough goals as well as the inclusion of sirolimus or everolimus [5,6]. These approaches must be weighed against rejection risk with lower tacrolimus exposure as well as additional side effects from sirolimus or everolimus. In addressing acute nephrotoxicity, the only established management options are either decreasing the tacrolimus dose or entirely holding for a period [4]. Both scenarios are associated with increased risk of rejection.

Basiliximab is an anti-CD25 monoclonal antibody used for the induction of immunosuppression by inhibiting IL-2 mediated T-cell proliferation [7]. The inhibitory effects of 20 mg basiliximab administered intravenously on post-operative day (POD) 0 and POD4 last 4–6 weeks, reduce rejection, and improve survival [8,9,10,11,12]. Additionally, basiliximab induction allows slower titration of tacrolimus to therapeutic goals, reducing the risk of nephrotoxicity without sacrificing the prevention of acute rejection [11,13].

Infrequently, basiliximab is used off-label outside of the induction period to facilitate further CNI reduction. Prior research on non-induction basiliximab (NIB) comprises four case series (three in heart recipients, one in lung recipients) and three case studies that summarize experiences with temporary tacrolimus cessation in the setting of acute kidney injury (AKI) [14,15,16,17,18,19,20]. Renal function recovery was noted in all patients in one case series while other cases reported no improvement [14,15,16,17,18,19,20]. In these reports, no comparator was utilized, and few addressed complications, including the occurrence of rejection or infection.

This study’s objective is to determine if the use of NIB allowed for renal recovery in cardiothoracic transplant recipients in the setting of AKI or of high risk for AKI. This study also aims to describe if renal recovery was sustained at 30 days as well as incidence of infection and acute rejection following the use of NIB for tacrolimus cessation.

## 2. Materials and Methods

This was a retrospective, single-center, descriptive study determined as exempt on 26 July 2023 by the Mayo Clinic Institutional Review Board (#23-005339). Electronic medical records of adult (≥18 years old) cardiothoracic transplant recipients who received basiliximab at POD7 or later between 1 January 2019 and 30 November 2023 were assessed for study inclusion. Basiliximab administration was classified as non-induction basiliximab (NIB) if given after POD7, outside of the standard induction dosing regimen of 20 mg basiliximab on POD0 and 4. Patients receiving NIB were classified into two cohorts, “temporary tacrolimus cessation” and “any tacrolimus reduction.” Temporary tacrolimus cessation was defined by the receipt of NIB to withdraw tacrolimus temporarily. Any tacrolimus reduction included patients in the temporary tacrolimus cessation group and patients that received NIB for temporary tacrolimus goal reduction without complete withdrawal. Acute kidney injury was defined as an increase in creatinine of 0.3 mg/dL or more from baseline within 48 h or an increase in creatinine of >1.5 times the baseline within 7 days, per KDIGO guidelines [21]. Patients were determined to be at high risk of AKI at the discretion of the treating physician’s assessment. Patients were excluded if they experienced concurrent rejection, if they received extrathoracic transplantation, or if no Minnesota research authorization was available.

The post-transplant immunosuppressive protocol at our facility for cardiothoracic transplant recipients is basiliximab (given on POD0 and POD4) for induction and maintenance immunosuppression with tacrolimus (goal trough range: 9–12 ng/mL), mycophenolate mofetil (adjusted to a white blood cell count of 4–8 × 10^9^/L), and prednisone. Highly sensitized patients were considered for induction with rabbit antithymocyte globulin.

The primary outcome was incidence of initial renal recovery after temporary tacrolimus cessation at 5 days after basiliximab administration compared to incidence of initial renal recovery at 5 days after diagnosis of AKI in the matched cohort. Renal recovery was defined as a creatinine less than or equal to the baseline creatinine plus 0.1 mg/dL without the need for dialysis. Baseline creatinine was defined as the median creatinine over the 30 days preceding diagnosis of AKI. Progression of renal recovery after AKI was assessed using the median creatinine through days 7, 14, and 30 less than or equal to the baseline creatinine plus 0.1 mg/dL without the need for dialysis. For patients receiving dialysis during a given period, creatinine measurements were omitted from analysis. The time to the initiation of dialysis after temporary tacrolimus cessation with NIB was recorded. The incidence of initial renal recovery, progression of renal recovery after AKI, and time to initiation of dialysis after any tacrolimus reduction with NIB were also recorded and assessed.

The safety of this intervention was assessed by the incidence of and time from AKI diagnosis to the first biopsy-proven acute rejection (BPAR) and treated infection up to one year. Acute cellular rejection for heart and lung recipients were graded per ISHLT guidelines and consensus reports, respectively [22,23,24,25]. The rates of the safety outcomes of patients that underwent temporary tacrolimus cessation were compared to those of the matched cohort.

### Statistical Methods

Statistical comparisons were performed between the temporary tacrolimus cessation group and the matched cohort. Categorical variables were described as frequencies and percentages and were compared between groups with Pearson’s Chi-Square analysis. Continuous variables were non-parametric in distribution; due to this and the limited sample size, they were described using medians (interquartile range [IQR]) and compared using the Mann–Whitney U test. The Kaplan–Meier method and Cox proportional hazards regression were used to describe the rate and time to dialysis, first treated infection, and first treated rejection. An alpha level of ≤0.05 was used to determine statistical significance. All analyses were performed using SAS version 9.4 software (SAS Institute, Inc.; Cary, NC, USA).

## 3. Results

In total, 76 patients were evaluated, and 20 were identified for potential inclusion, having received basiliximab outside of the induction period, as summarized in Figure 1. One patient was excluded, having received NIB in the setting of temporary tacrolimus cessation for neurotoxicity and expired 16 days later. Twelve patients received NIB for temporary tacrolimus cessation in the setting of AKI. A total of 19 patients received NIB for any tacrolimus reduction in the setting of AKI or high risk for AKI. Patients receiving temporary tacrolimus cessation were matched 1:3 to a cohort of 36 patients that experienced AKI but did not receive NIB.

Baseline characteristics are described in Table 1. In the temporary tacrolimus cessation group (n = 12), the median duration of temporary tacrolimus cessation with NIB was 5 [4,5,6] days. An example timeline of temporary tacrolimus cessation is shown in Figure 2. Twenty-nine (53%) patients had at least one year of follow-up data available, and forty-three (78%) patients had at least six months of follow-up data available. A total of 9 (75%) patients in the temporary tacrolimus cessation group had AKI stage 2 or higher, while 31 (86%) patients in the matched cohort had AKI stage 1.

Of the 12 patients with temporary tacrolimus cessation, concomitant nephrotoxins were used in 4 (33%) patients, vancomycin (n = 2) and contrast media (n = 2). The matched cohort included 9 (25%) patients who received concomitant nephrotoxins, vancomycin (n = 7) and NSAIDs (n = 2). Of the 19 patients with any tacrolimus reduction, concomitant nephrotoxins were used in 8 (42%) patients, vancomycin (n = 6) and contrast media (n = 2).

The 12 patients undergoing temporary tacrolimus cessation had other documented potential physiological causes of AKI, including post-operative AKI (n = 3), hypervolemia (n = 6), hypovolemia (n = 5), and shock state (n = 2). Ten (28%) patients in the matched cohort had other documented potential physiological causes of AKI, including hypervolemia (n = 5), hypovolemia (n = 3), shock state (n = 3), and post-operative AKI (n = 2). All patients undergoing any tacrolimus reduction had other documented potential physiological causes of AKI, including post-operative AKI (n = 9), hypervolemia (n = 7), hypovolemia (n = 6), and shock state (n = 2). In total, 3 (25%) of 12 patients experienced changes to immunosuppression regimens after undergoing temporary tacrolimus cessation (Table 2). Notable regimen changes were tacrolimus to cyclosporine (n = 1) or tacrolimus to tacrolimus plus sirolimus (n = 2). Four (36%) patients that remained on tacrolimus (n = 11) had goal levels reduced after undergoing temporary tacrolimus cessation. In the matched cohort, one (3%) patient was changed from tacrolimus plus sirolimus to sirolimus only, and six (17%) had tacrolimus goal levels reduced. None of the additional seven patients in the any tacrolimus reduction group experienced changes to immunosuppression regimens.

Median tacrolimus levels in the week following NIB for any tacrolimus reduction were recorded and reported in Table 2. Median tacrolimus levels in the week following NIB for temporary tacrolimus cessation and median tacrolimus levels in the week following the equivalent period for the matched cohort were also recorded and reported in Table 2. In the temporary tacrolimus cessation group, 4 (33%) patients had undetectable levels when tacrolimus or alternative therapy was initiated. In the matched cohort, only two (6%) patients had a tacrolimus level <4 ng/mL. Seventeen (89%) patients had tacrolimus levels <4 ng/mL during the period of any tacrolimus reduction.

Renal recovery outcomes are summarized in Table 2. In the temporary tacrolimus cessation group, seven (58%) patients achieved initial renal recovery. This was not significantly higher than the 15 (42%) patients in the matched cohort with initial renal recovery at 5 days after AKI diagnosis (*p* = 0.316). Progression of renal recovery after temporary tacrolimus cessation was assessed on days 7, 14, and 30, where 0, 3 (25%), and 5 (42%) patients had sustained renal recovery, respectively. In the matched cohort, sustained renal recovery on days 7, 14, and 30 was seen in 13 (36%), 15 (42%), and 15 (42%) patients, respectively. The only significant difference was sustained renal recovery on day 7 after AKI (*p* = 0.015), while sustained renal recovery on days 14 and 30 was not significantly different (*p* = 0.30 and >0.99, respectively).

After any tacrolimus reduction, eight (42%) patients achieved initial renal recovery. Progression of renal recovery was assessed on days 7, 14, and 30, where 0, 2 (16%), and 6 (32%) patients sustained renal recovery. Six (32%) patients were dialysis-dependent within the first year. Notably, 5 of the 11 patients who did not experience renal recovery were on continuous renal replacement therapy (CRRT) throughout the reduction period.

Rejection and infection outcomes are summarized in Table 3. Among the 12 patients in the temporary tacrolimus cessation group, the 2 patients (17%) with at least one episode of BPAR after NIB were not statistically significantly different from the 7 (19%) patients in the matched cohort. Median time from AKI to first treated rejection for the matched cohort was 52 [48–185] days. Rejections treated in the matched cohort included grade 2R acute cellular (n = 3) and antibody-mediated (n = 1) rejections in heart recipients and an acute cellular A1 symptomatic and A2 (n = 2) rejections in lung recipients.

After any tacrolimus reduction, seven (37%) patients had at least one episode of BPAR and two (11%) required treatment. The two patients requiring treatment for BPAR were lung recipients with grade A2 rejection at 94 and 209 days from NIB. There were no episodes of antibody-mediated rejection after any tacrolimus reduction.

In the temporary tacrolimus cessation group, 9 (75%) patients had at least one treated infection with a median time to first treated infection of 50 [30–109] days. This was not statistically significantly different from the 18 (50%) patients with at least one treated infection in the matched cohort with median time to first treated infection of 67 [26–91] days (HR: 1.52; 95% CI 0.68–3.38; *p*-value = 0.31). A total of 15 (79%) patients had at least one treated infection after any tacrolimus reduction and a median time to first treated infection of 52 [33–123] days.

## 4. Discussion

This evaluation of temporary tacrolimus cessation using basiliximab outside of the induction period demonstrated potential benefit in facilitating renal recovery. Despite having higher stages of AKI, patients that received NIB for temporary tacrolimus cessation had similar renal function recovery compared to the matched cohort. There was no difference in rejection or infection rates.

Our study is the first to compare outcomes of NIB for temporary tacrolimus cessation in the setting of AKI to a matched control group. In this group, 7 (58%) of 12 patients achieved renal recovery after temporary tacrolimus cessation with NIB for AKI, compared to 15 (42%) of 32 patients with AKI that did not receive NIB. The median duration of tacrolimus cessation was 5 days, which was shorter than previously reported experiences of 6 days to 18 months [14,15,16,17,18,19,20]. Thirty days after AKI, we observed no difference in renal recovery between the group receiving temporary tacrolimus cessation and the matched cohort. More patients went on to need dialysis after receiving NIB for temporary tacrolimus cessation. This could be partially due to the severity of renal dysfunction observed in the group receiving NIB.

Multiple approaches have been used to minimize nephrotoxic impacts of tacrolimus. In renal transplant, studies have been conducted to determine the lowest CNI trough levels possible to improve renal function while still retaining adequate rejection prophylaxis [28]. Delaying CNI initiation with induction agents has also been studied. Induction with basiliximab compared to steroids alone demonstrated similar outcomes in rejection and infections, with a trend toward less rejection and improved renal function in those receiving basiliximab. Tacrolimus initiation occurred on POD 3–7 with slow titration to goal levels at 2–4 weeks, compared to steroid-only induction with tacrolimus, which started at POD 1, and faster titration to therapeutic levels at 4–14 days [10,11,13]. A novel approach has been the use of IL-2 inhibition outside of the induction phase to facilitate temporary tacrolimus cessation or tacrolimus reduction.

In the early 2000s, two studies attempted to avoid the use of CNIs after kidney transplant with daclizumab, an IL-2 inhibitor [29,30]. While both studies noted improvement in renal function, this was complicated by 12-month rejection rates of 53% to 70%, raising concern for the immunologic efficacy of long-term IL-2 inhibition [29,30]. A reason for this may lie in mechanistic differences between tacrolimus and IL-2 inhibitors. Murine model assessment has shown that CD8 cells can initiate the cell cycle in the absence of IL-2 [31], possibly explaining the increased rejection rate seen in the renal transplant studies and suggesting the need for an immunologic target farther upstream than IL-2 for maintenance immunosuppression. Inhibition of calcineurin inhibits production of CD40L, IL-2, IL-4, IL-17, and other cytokines associated with T cell activation [32,33]. Interestingly, CNIs have been associated with a possible reduction in IL-15 in murine models [34]. Importantly, IL-15 has shown potential to mediate a pathway that bypasses IL-2 inhibition and may precipitate acute rejection [32,35]. This is likely due to the constitutively expressed β and γc subunits, which are shared by IL-15 and IL-2 [32,36]. As basiliximab binds to the α subunit of the IL-2 receptor (which is specific for IL-2 and not IL-15), it has no antagonistic activity against IL-15 receptors, and T cells that have undergone signal 1 and 2 activation can plausibly be stimulated to proliferate. Inhibiting progression of signals 1 and 2 through calcineurin inhibition may prevent sustained expansion to IL-15 stimulation. These nuance differences may underline the increased rejection risk observed with long-term IL-2 inhibition and complete CNI avoidance.

To date, IL-2 inhibition, primarily basiliximab, has been used as a maintenance agent for renal recovery in five case series (three in heart recipients, two in lung recipients) and three cases studies [14,15,16,17,18,19,20,37]. When used to reduce tacrolimus exposure, target levels were 2–4 ng/mL, and basiliximab was used for up to 12 months [19,37]. When used for the temporary cessation of CNI, IL-2 inhibition was used to hold tacrolimus between 6 days and 18 months [12,13,14,15,16,17,18]. In these experiences, 36 out of 48 patients saw stabilization or improvement of renal function at the end of the observed period [14,15,16,17,18,19,20]. The most common NIB regimen across these studies was a single dose followed by another dose four days later when needed, similar to our approach [14,15,16,17,18,19,20]. When further NIB was used, regimens varied from weekly doses to repeat doses every 8 weeks [14,15,16,17,18,19,20].

The impact of any tacrolimus reduction with NIB after the observed period is addressed in one case study and one case series. The case study by Anselm et al. does not report baseline creatinine but shows improved serum creatinine at 24 months, with the patient receiving 18 months of basiliximab followed by transition to sirolimus, which they remained on [16]. The case series by Cantarovich et al. observed an approximate 10% increase in serum creatinine 1 week after the resumption of CNIs [20]. Our study adds to the limited assessment of renal function with an assessment of creatinine up to 30 days after the use of NIB. We saw a small numerical reduction in the median creatinine through 30 days after AKI diagnosis compared to immediately after the use of NIB for temporary tacrolimus cessation; however, numerically, less patients met criteria for renal recovery through day 30 after AKI (n = 5) than at the end of tacrolimus cessation. Initial renal recovery after temporary tacrolimus cessation was defined by a single creatinine level while our secondary assessment analyzed median creatinine levels over a period of time to assess trends, which may explain this observation. Through these trends, we observed a continuous reduction in median creatinine levels after temporary tacrolimus cessation with NIB.

Consideration must be taken when observing the difference in changes to immunosuppression regimens as described in Table 2. The significance (*p* = 0.02) of these changes was primarily driven by the addition of sirolimus in two patients in the tacrolimus cessation group (*p* = 0.01). In the context of severe AKI, changes to immunosuppression to facilitate renal recovery falls within our standard practice. Non-induction basiliximab’s role in transitioning to a new maintenance regimen is novel.

Incidence of infection or rejection beyond the use of NIB for temporary tacrolimus cessation is mentioned in one case series. Eiting et al. observed 25 (96%) lung recipients with an infection by 6 months and 10 (38%) experienced either BPAR or suspected rejection [13]. No tacrolimus levels were recorded, and basiliximab was used for a mean of 1.5 doses given at least two weeks apart [15].

The lack of recorded tacrolimus levels is notable because of the concerns of increased rejection risk based on mechanistic differences between tacrolimus and basiliximab. As mentioned, most of our patients in the any tacrolimus reduction group had trough levels <4 ng/mL during the reduction period. In one kidney transplant study, tacrolimus levels <4 ng/mL were associated with a 6.33 fold increased risk for rejection within 12 months of transplant [38]. Another kidney transplant study that attempted complete withdrawal of tacrolimus without any replacement therapy reported that rejection occurred in 4 out of 13 patients at either 15 or 78 days after the withdrawal [39]. Together, this suggests tacrolimus cannot be maintained at these subtherapeutic levels beyond a few weeks without increasing rejection risk.

When comparing temporary tacrolimus cessation with NIB to a matched cohort without NIB, the safety of temporary tacrolimus cessation is reinforced when noting that all but one patient experiencing rejection in the matched cohort had tacrolimus levels >4 ng/mL. The only patient that had a tacrolimus level <4 ng/mL was transitioned to sirolimus. Therefore, the use of NIB for a short period of tacrolimus cessation, approximately 5 days in our study, did not appear to impact the rejection risk in the long term.

Reported infection rates associated with basiliximab induction therapy range from 6 to 41% [9,10,11,13]. When using NIB for any tacrolimus reduction, a higher incidence of treated infection was seen of 79% in our study. This was lower than what has been previously reported for NIB, with 96% of patients in one case series having an infection by 6 months [15]. Though not a statistically significant difference compared to 50% in the matched cohort, the 75% infection rate in the temporary tacrolimus cessation group may signal a clinically significant difference that should be noted when considering the use of NIB. This could be confounded by the acuity of individuals receiving NIB, with all patients receiving NIB being hospitalized and having higher stages of AKI, and more patients requiring vasopressor support. Further research into infection risk with NIB with larger cohorts can help clarify possible confounders.

Limitations inherent to a retrospective, single-center chart review are present in our study. These include biases in patient selection and clinical decision making. The use of NIB for any tacrolimus reduction is not protocol-based, further leading to potential for selection bias. Additionally, baseline creatinine was defined as the 30 days prior to AKI diagnosis. This occasionally included creatinine values prior to transplant and led to our assessment of renal function being more conservative and may underrepresent true benefits. Patients that received renal replacement therapy during any tacrolimus reduction with NIB may have had artificially lowered creatinine levels immediately after the reduction period followed by an observed rise in creatinine as they approached their new baseline, further contributing to our conservative estimate. The temporary tacrolimus cessation group had more severe AKI compared to the matched cohort; this is possibly a reflection of provider selection bias for the intervention and a limitation of a retrospective study. Propensity matching to reduce selection bias and increase result validity was considered but determined not feasible with the small sample size.

This study adds value to the transplant pharmacotherapy literature by describing a unique approach in the use of NIB for renal recovery in heart and lung transplant recipients. Though no standardized protocol was available with the novelty of this approach, this study can serve as a foundation for protocol development to facilitate a standardized approach. Intriguingly, basiliximab along with co-stimulation blocker belatacept may pose additional benefit given the mechanism of action being farther upstream in comparison to basiliximab alone, and could be worth further investigation [40,41]. Continued research into the appropriate duration of NIB, approaches to and the duration of tacrolimus reduction with or without concomitant therapies, and long-term outcomes of these practices is still needed to optimally apply approaches.

## 5. Conclusions

The use of NIB for temporary tacrolimus cessation can allow for potential renal recovery after an AKI. Based on this single-center experience with a limited sample size, this approach does not appear to significantly increase rejection risk but does potentially increase infection risk in the long term.

## Figures and Tables

**Figure 1 medicines-12-00022-f001:**
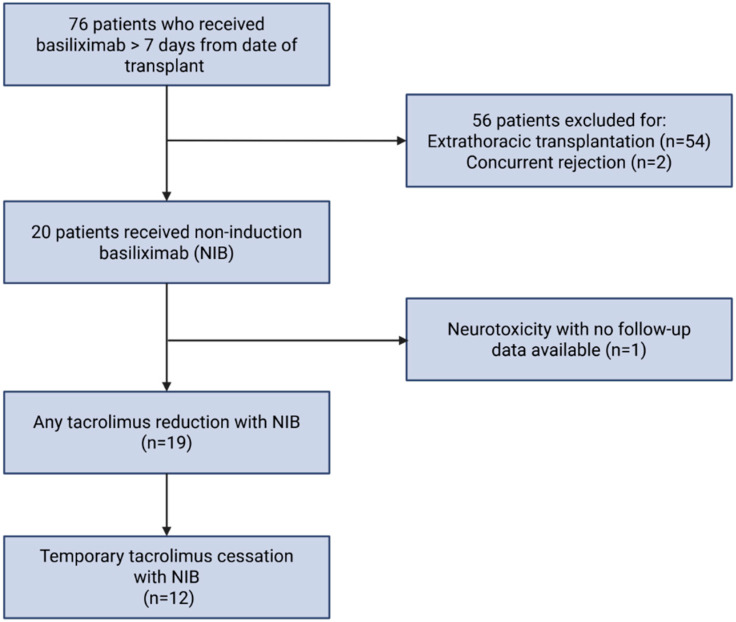
Patient identification [26].

**Figure 2 medicines-12-00022-f002:**
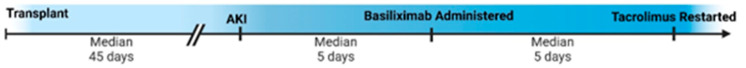
Median timeline of patients receiving NIB for temporary tacrolimus cessation [27].

**Table 1 medicines-12-00022-t001:** Baseline demographics.

	Any Tacrolimus Reduction (n = 19)	Tacrolimus Cessation Group (n = 12)	Matched Comparison Group (n = 36)
Age, Median (Q1, Q3)	65 [53–69]	66 [53–69]	63 [57–67]
Birth Sex (Male)	14 (74%)	7 (58%)	19 (53%)
Race			
White	11 (58%)	9 (75%)	30 (83%)
Black or African Ancestry	6 (32%)	2 (17%)	2 (6%)
Asian	1 (5%)	1 (8%)	2 (6%)
American Indian/Alaskan Native	1 (5%)	0	1 (3%)
Other	0	0	1 (3%)
Diabetes	8 (42%)	6 (50%)	12 (33%)
Hypertension	8 (42%)	4 (33%)	17 (47%)
Hyperlipidemia	13 (68%)	9 (75%)	21 (58%)
Chronic Kidney Disease	9 (47%)	3 (25%)	17 (47%)
Transplant Type			
Heart	6 (32%)	2 (17%)	10 (28%)
Lung	12 (63%)	10 (83%)	25 (69%)
Simultaneous Heart and Lung	1 (5%)	0	1 (3%)
Induction			
Steroids only	5 (26%)	3 (25%)	9 (25%)
Basiliximab	12 (63%)	8 (67%)	23 (64%)
Rabbit antithymocyte globulin	2 (11%)	1 (8%)	4 (11%)
History of biopsy-proven rejection prior to tacrolimus reduction	4 (21%)	4 (33%)	17 (47%)
Hospitalization at event	19 (100%)	12 (100%)	30 (83%)
Baseline SCr (mg/dL)	1.3 [1.2–1.6]	1.3 [1.1–2.1]	1.5 [1.1–1.7]
SCr at diagnosis of AKI (mg/dL)	1.9 [1.6–2.2]	2.0 [1.6–2.6]	1.9 [1.5–2.3]
SCr at administration of NIB (mg/d) ^a^	2.2 [1.9–4.2]	2.7 [1.9–4.4]	-
Peak SCr (mg/dL)	2.4 [2.2–4.1]	2.8 [2.2–4.4]	2.1 [1.9–2.7]
AKI Stage			
1	4 (21%)	3(25%)	31 (86%)
2	4 (21%)	4 (33%)	4 (11%)
3	10 (53%)	5 (42%)	1 (3%)
Concomitant nephrotoxins present	8 (42%)	4 (33%)	9 (25%)
Concomitant vasopressors present	10 (53%)	5 (42%)	3 (8%)
Renal replacement required during reduction	6 (32%)	1 (8%)	1 (3%)
Other physiological causes of AKI present	19 (100%)	12 (100%)	10 (28%)
Immunosuppression before NIB			
Tacrolimus, MMF, Prednisone	14 (74%)	9 (75%)	28 (78%)
Tacrolimus, Prednisone	5 (26%)	3 (25%)	7 (19%)
Tacrolimus, Sirolimus, MMF, Prednisone	0	0	1 (3%)
Median tacrolimus level week prior to AKI (ng/mL)	8 [4.5–10.1]	8.1 [5.8–10.2]	9.1 [7.5–11]
Time from transplant to AKI (months)	0.3 [0.05–4.0]	1.5 [0.3–5.8]	2.6 [1.2–7.7]

Data presented as n (%) or median [IQR]. (^a^) Creatinine levels taken while receiving renal replacement therapy were omitted (n = 5). MMF: mycophenolate mofetil; SCr: serum creatinine; AKI: acute kidney injury; NIB: non-induction basiliximab.

**Table 2 medicines-12-00022-t002:** Renal recovery from tacrolimus reduction.

	Any Tacrolimus Reduction (n = 19)	Tacrolimus Cessation Group (n = 12)	Matched Comparison Group (n = 36)	*p*-Value ^a^
Number of doses of NIB received				-
One dose	10 (53%)	3 (25%)	-	
Two doses	8 (42%)	8 (67%)	-	
Three doses	1 (5%)	1 (8%)	-	
Renal recovery at end of tacrolimus reduction	8 (42%)	7 (58%)	15 (42%)	0.32
Required dialysis at end of tacrolimus reduction	4 (21%)	1 (8%)	0	0.08
SCr (mg/dL) at end of reduction period ^b c^	1.7 [1.5–2.4]	1.5 [1.3–2.2]	1.6 [1.3–1.9]	0.82
Median SCr (mg/dL) days 0–7 post AKI ^b^	2.2 [1.9–3.2]	2.0 [1.8–3.4]	1.8 [1.3–2.0]	0.041
Median SCr (mg/dL) days 8–14 post AKI ^b^	1.8 [1.4–2.3]	1.5 [1.4–2.4]	1.5 [1.2–2.1]	0.39
Median SCr (mg/dL) days 15–30 post AKI ^b^	1.6 [1.3–1.9]	1.4 [1.3–1.8]	1.5 [1.3–2.0]	0.98
Renal recovery at 30 days post-AKI	6 (32%)	5 (42%)	15 (42%)	>0.99
Dialysis dependent within 30 days	4 (21%)	1 (8%)	1 (3%)	0.40
Median tacrolimus level one week following NIB (ng/mL)	7.0 [4.9–7.7]	5.6 [4.3–7.6]	8.8 [7.6–9.8]	0.001
Went on to be dialysis-dependent within one year	6 (32%)	3 (25%)	2 (6%)	0.092
Changes to immunosuppression regimen		3 (25%)	1 (3%)	0.02
Change of tacrolimus to cyclosporine	1 (5%)	1 (8%)	0	0.08
Addition of sirolimus	2 (11%)	2 (17%)	0	0.01
Changed to sirolimus monotherapy	0	0	1 (3%)	0.56
Reduction in tacrolimus trough goal ^d^	11 (61%)	4 (36%)	6 (17%)	0.18

Data presented as n (%) or median [IQR]. (^a^) Statistical comparison of temporary tacrolimus cessation and matched comparison group. (^b^) Creatinine levels taken while receiving renal replacement therapy were omitted (n = 5). (^c^) Creatinine at 5 days after basiliximab administration was used for the assessment of patients who received NIB. In the matched cohort, creatinine at 5 days after AKI diagnosis was used for assessment. (^d^) Patients changed to cyclosporine and sirolimus were omitted from this assessment. SCr = serum creatinine.

**Table 3 medicines-12-00022-t003:** Safety outcomes from tacrolimus reduction.

	Any Tacrolimus Reduction (n = 19)	Tacrolimus Cessation Group (n = 12)	Matched Comparison Group (n = 36)	*p*-Value ^a^
Treated biopsy-proven rejection after AKI	2 (11%)	2 (17%)	7 (19%)	0.80
Any biopsy-proven rejection after AKI	7 (37%)	4 (33%)	14 (39%)	>0.99
Patients with at least one treated infection after tacrolimus reduction ^b^	15 (79%)	9 (75%)	18 (50%)	0.32
Total number of treated infections	32	20	53	-
CMV	1	0	2	-
EBV	2	2	1	-
Fungal	1	1	2	-
Bacterial	25	16	35	-
Viral	4	2	15	-

Data presented as n (%) or median [IQR]. (^a^) Statistical comparison of temporary tacrolimus cessation and matched comparison group. (^b^) Includes patients treated for coinfection. AKI: acute kidney injury.

## Data Availability

The data underlying this article will be shared on reasonable request to the corresponding author.

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
