# Peer review of "Non-Induction Basiliximab to Facilitate Renal Recovery via Temporary Tacrolimus Cessation in Cardiothoracic Transplant Patients"

_medicines, 2025, doi:10.3390/medicines12030022_

Round 1

Reviewer 1 Report

Comments and Suggestions for Authors

Thank you for reviewing this interesting manuscript. Immunosuppressive agents used in organ transplantation are essential, but they can cause many side effects. Tacrolimus plays a crucial role in all organ transplants, although its renal side effects are well-documented. It is not indispensable in kidney transplantation, but after heart or lung transplants—where patients are at risk of developing potential AKI due to long surgical procedures—temporary cessation of tacrolimus, which can damage the kidneys, should be considered. The authors retrospectively examined the outcome of temporarily stopping tacrolimus after basiliximab induction. The procedure was found to be safe and did not increase the risk of rejection. The authors' findings are noteworthy. The main limitation of their study is that it is a single-center experience with a very small number of cases, so this should be more clearly emphasized in the conclusion. Additionally, limitations include that it was not a prospective study, the follow-up was relatively short, and its long-term effects on chronic rejection were not evaluated.

Author Response

Thank you for reviewing this interesting manuscript. Immunosuppressive agents used in organ transplantation are essential, but they can cause many side effects. Tacrolimus plays a crucial role in all organ transplants, although its renal side effects are well-documented. It is not indispensable in kidney transplantation, but after heart or lung transplants—where patients are at risk of developing potential AKI due to long surgical procedures—temporary cessation of tacrolimus, which can damage the kidneys, should be considered. The authors retrospectively examined the outcome of temporarily stopping tacrolimus after basiliximab induction. The procedure was found to be safe and did not increase the risk of rejection. The authors' findings are noteworthy. The main limitation of their study is that it is a single-center experience with a very small number of cases, so this should be more clearly emphasized in the conclusion. Additionally, limitations include that it was not a prospective study, the follow-up was relatively short, and its long-term effects on chronic rejection were not evaluated.

We agree with the points raised by the reviewer. Lines 357-359 in the “Conclusion” have been modified to further emphasize our limitations of being a single-center with limited sample size. Though outside the scope of this study in its current form, we agree long-term effects should be assessed as patients are farther out from this intervention. 

Reviewer 2 Report

Comments and Suggestions for Authors

This manuscript is a well-written, focused, and clinically relevant retrospective study exploring the utility of basiliximab administered outside the induction period (non-induction basiliximab, NIB) to temporarily discontinue tacrolimus in cardiothoracic transplant patients experiencing or at high risk for acute kidney injury (AKI).  The study adds to the limited existing literature by including a matched control group and assessing safety and renal recovery outcomes.

  1. This is the first study to compare outcomes of NIB for tacrolimus cessation against a matched cohort, filling an important gap in transplant pharmacotherapy literature.  The discussion could be strengthened by explicitly comparing outcomes with prior case series (e.g., sample sizes, follow-up duration, dosing regimens).  The authors should consider emphasizing the potential translational implications for immunosuppression protocols in centers managing high-risk transplant patients.
  2. The authors provide a thorough immunologic rationale for temporary tacrolimus cessation with NIB.  While the discussion of IL-2/IL-15 mechanisms is interesting, consider shortening or summarizing it more concisely to maintain clinical focus.

Author Response

1.This is the first study to compare outcomes of NIB for tacrolimus cessation against a matched cohort, filling an important gap in transplant pharmacotherapy literature.  The discussion could be strengthened by explicitly comparing outcomes with prior case series (e.g., sample sizes, follow-up duration, dosing regimens).  The authors should consider emphasizing the potential translational implications for immunosuppression protocols in centers managing high-risk transplant patients.

The commentary to strengthen the discussion by making comparisons more explicit is an appreciated point. Rephrasing and clarifying phrases have been added to lines 230-232, 278-277, 283-285. Protocol information has been added to line 274-277.

A limitation of this study was the lack of a standardized protocol given the novelty of this intervention. This can serve as a foundation for protocol development to facilitate a standardized approach. This statement has been added to line 347-349.

2.The authors provide a thorough immunologic rationale for temporary tacrolimus cessation with NIB.  While the discussion of IL-2/IL-15 mechanisms is interesting, consider shortening or summarizing it more concisely to maintain clinical focus.

If the editors do not find this section of value, the authors are OK removing this from the discussion. This content was added because mechanistic granularity for the outcomes observed in lines 248-251 has not been expressed in previous literature and acknowledging the multimodal suppression of T-cells provided by tacrolimus as compared to the specific, single target mechanism provided by basiliximab is worthy of comment. 

Reviewer 3 Report

Comments and Suggestions for Authors

Tanner A Melton et al. investigated the use of non-induction basiliximab combined with temporary tacrolimus cessation in heart and lung transplant recipients to prevent or mitigate tacrolimus-induced acute kidney injury (AKI). The study holds great clinical significance, addressing a clinically relevant issue and proposing a potential strategy for managing acute kidney injury (AKI) in cardiothoracic transplant patients. However, there still exist several concerns seriously limiting the strength and interpretability of the findings:

  1. The rationale of this study is to reduce the nephrotoxic side effects of tacrolimus. The introduction should elaborate on the currently available strategies or methods to mitigate tacrolimus-induced nephrotoxicity, including their respective advantages and disadvantages.
  2. Please clarify whether ethical approval was obtained for this study. The manuscript should include an ethics statement along with the relevant ethics approval number.
  3. Basiliximab is most commonly used for induction therapy or rejection treatment. When used for early maintenance therapy, does this fall within the approved indications for use?
  4. Since both heart and lung transplant recipients were included, subgroup analyses for heart and lung transplant recipients are suggested.
  5. Some information in the Methods section is redundantly described (e.g., lines 100-103) and should be streamlined for clarity.
  6. The Kruskal-Wallis test is a non-parametric test used for comparing multiple independent samples based on a single factor. However, in this study, patients in the “any tacrolimus reduction” group and the “temporary tacrolimus cessation” group are not independent samples due to overlapping membership. Therefore, the use of the Kruskal-Wallis test is not appropriate. The Mann-Whitney U test would be more suitable in this context.
  7. The definition of patients at high risk for AKI (line 123) should be clearly stated.
  8. A comparative analysis of baseline characteristics between the experimental and control groups is required. P-values should be provided, as underlying conditions such as chronic kidney disease (CKD) may increase the risk of acute kidney injury (AKI).
  9. The description of the results should be consistent with the tables (e.g., line 151 states 28%, whereas Table 1 reports 29%).
  10. It is recommended to replace the comma in the interquartile range (IQR) with a hyphen when reporting median [IQR] values. For example, "52 [48, 185]" is formatted as "52 [48-185]".
  11. Information regarding the dosage or AUC (if available) of mycophenolate mofetil (MMF) should be included and compared between groups, as MMF dosage may also influence rejection outcomes.
  12. To better evaluate the safety and efficacy of NIB treatment, changes in cardiac and pulmonary function indicators should be reported, not only the incidence of rejection.
  13. In the Tacrolimus Cessation Group, the median duration of NIB treatment was 5 days. In Table 2, does the reported “Median tacrolimus level one week following NIB” reflect a potentially unstable drug concentration level as it need time to reach stable levels?
  14. In the Tacrolimus Cessation Group, the median duration of intervention was less than a week, but the follow-up period extended up to one year. Please clarify how renal function recovery was attributed to this short-term intervention, and whether any renoprotective treatments were administered during the follow-up period.
  15. Please carefully review the manuscript for clarity of language and revise the formatting of tables as needed.

Author Response

1.The rationale of this study is to reduce the nephrotoxic side effects of tacrolimus. The introduction should elaborate on the currently available strategies or methods to mitigate tacrolimus-induced nephrotoxicity, including their respective advantages and disadvantages.

Lines 50-56 were modified for clarity. Information for mitigation of chronic nephrotoxicity was briefly mentioned. The focus of this study was to address acute nephrotoxicity, of which the only established methods are reducing or holding tacrolimus. 

2.Please clarify whether ethical approval was obtained for this study. The manuscript should include an ethics statement along with the relevant ethics approval number.

This study was deemed exempt by the local IRB. This was recorded on lines 75-76. An ethics statement was added per editor request on lines 368-369 as well under the title “Institutional Review Board Statement.” This has been changed to “Ethics Statement” for added clarity. The study IRB number and date are #23-005339 and 7/26/2023, respectively. This information was added to lines 75-76 and line 369-370.

3.Basiliximab is most commonly used for induction therapy or rejection treatment. When used for early maintenance therapy, does this fall within the approved indications for use?

This is not within approved indications and is used off-label in this study. Wording on line 63 has been adjusted for clarity.

4.Since both heart and lung transplant recipients were included, subgroup analyses for heart and lung transplant recipients are suggested.

Statistical comparisons were done between the tacrolimus cessation group and the matched cohort only. Given the small number of heart transplants (n=2) in the tacrolimus cessation group, it was determined that a subgroup analysis would not provide meaningful results. 

5.Some information in the Methods section is redundantly described (e.g., lines 100-103) and should be streamlined for clarity.

The authors appreciate the commentary to help streamline this manuscript. The duplicative information was consolidated and is now lines 106-108 as “Incidence of initial renal recovery, progression of renal recovery after AKI, and time to initiation of dialysis after any tacrolimus reduction with NIB were also recorded and assessed.”

6.The Kruskal-Wallis test is a non-parametric test used for comparing multiple independent samples based on a single factor. However, in this study, patients in the “any tacrolimus reduction” group and the “temporary tacrolimus cessation” group are not independent samples due to overlapping membership. Therefore, the use of the Kruskal-Wallis test is not appropriate. The Mann-Whitney U test would be more suitable in this context

The groups of “any tacrolimus reduction” and “temporary tacrolimus cessation” are never compared statistically. A clarifying statement has been added to lines 114-115, and a solid line has been included in the tables for visual clarity.

“Statistical comparisons were performed between the temporary tacrolimus cessation group and the matched cohort.”

7.The definition of patients at high risk for AKI (line 123) should be clearly stated.

The clarifying statement “ Patients were determined to be high risk of AKI at the discretion of the treating physician’s assessment.” was added to lines 88-89.

8.A comparative analysis of baseline characteristics between the experimental and control groups is required. P-values should be provided, as underlying conditions such as chronic kidney disease (CKD) may increase the risk of acute kidney injury (AKI).

P-values were not included as the STROBE guidelines recommend against including inferential measures or significance tests in the description of baseline variables. The following reference “The Strengthening the Reporting of Observational Studies in Epidemiology (STROBE) statement: guidelines for reporting observational studies”. This can be included in the manuscript if desired. If the editorial team desires p-values to be added, this can be accommodated.

9.The description of the results should be consistent with the tables (e.g., line 151 states 28%, whereas Table 1 reports 29%).

The authors are appreciative of the reviewer’s keen eye. The 29% has been changed to 28%.

10.It is recommended to replace the comma in the interquartile range (IQR) with a hyphen when reporting median [IQR] values. For example, "52 [48, 185]" is formatted as "52 [48-185]".

These adjustments have been made, and the recommendation is appreciated. 

11.Information regarding the dosage or AUC (if available) of mycophenolate mofetil (MMF) should be included and compared between groups, as MMF dosage may also influence rejection outcomes.

The reviewer raises excellent points with this commentary. Unfortunately, our transplant center does not routinely measure mycophenolic acid AUC. Specific mycophenolate dose data was not collected as our protocol adjusts mycophenolate dose based on white blood cell and lymphocyte count. Because we did not find a difference in rejection rate between those who received NIB and the matched comparator group, the added mycophenolate dosing data is unlikely to add significant value. If this data is desired, an extension is requested to provide time for collection. 

12.To better evaluate the safety and efficacy of NIB treatment, changes in cardiac and pulmonary function indicators should be reported, not only the incidence of rejection.

Data on pulmonary and cardiac function were not collected. Clinically significant decline in cardiac and pulmonary function triggers a rejection work up. Given the primary outcome was to evaluate the effectiveness of NIB on AKI resolution, reporting rejection as a secondary safety outcome of immunosuppression modification is sufficient. If these additional data points are desired, an extension is requested to provide time for collection.

13.In the Tacrolimus Cessation Group, the median duration of NIB treatment was 5 days. In Table 2, does the reported “Median tacrolimus level one week following NIB” reflect a potentially unstable drug concentration level as it need time to reach stable levels?

This time point was used to reflect the variability of time to start tacrolimus, diversity in clinical practice, and assess for trends in renal recovery.

14.In the Tacrolimus Cessation Group, the median duration of intervention was less than a week, but the follow-up period extended up to one year. Please clarify how renal function recovery was attributed to this short-term intervention, and whether any renoprotective treatments were administered during the follow-up period.

AKI is known to impact acute morbidity and mortality in addition to long term outcomes such as progression of underlying CKD and renal failure requiring dialysis in severe cases. The intervention of NIB with CNI cessation to reduce additional kidney injury from tacrolimus is in addition to standards of care for managing AKI (reducing/eliminating nephrotoxins, fluid management, etc). We cannot draw the conclusion that NIB with CNI cessation is responsible for long term kidney function outcomes for these patients. However, reporting these outcomes in addition to the more direct short-term effects on kidney function are a strength of this study. It provides insight into if this major intervention could contribute to a difference long term kidney function. Considering >40% of the NIB and CNI cessation group had an AKI stage 3, long term kidney function outcomes are worthy of report, even if differences are not statistically significant. 

15.Please carefully review the manuscript for clarity of language and revise the formatting of tables as needed

We appreciate the reviewer’s comments that help strengthen this manuscript. 

Reviewer 4 Report

Comments and Suggestions for Authors

In the study by Tanner A. Melton, the authors investigated “Non-induction Basiliximab to Facilitate Renal Recovery via Temporary Tacrolimus Cessation in Cardiothoracic Transplant Patients.”

Calcineurin inhibitors (CNIs) play a pivotal role in transplant medicine and are employed in nearly all thoracic transplant cases. However, due to the long-term nature of post-transplant follow-up, CNI-associated nephrotoxicity remains a critical clinical issue. This study is intriguing in that it explores the potential renoprotective effect of temporary CNI discontinuation through the administration of basiliximab at the onset of acute kidney injury (AKI).

However, several limitations should be addressed:

Concern #1
The most significant limitation of this study is the small sample size. Although matching was used to select control subjects, the choice of matching parameters is inherently subject to investigator bias, making it difficult to fully eliminate selection bias. From a clinical standpoint, the sample size is too limited to draw definitive conclusions regarding the efficacy of the basiliximab-based regimen on renal function in the setting of AKI. The authors are encouraged to either substantially increase the sample size or justify the adequacy of the current sample through power analysis or similar methodology.

Concern #2
The definition of AKI used in the study appears to be insufficiently rigorous. A transient increase in serum creatinine (e.g., 0.3 mg/dL) can occur with dehydration and may resolve with simple hydration, which would not reflect true renal injury. The authors should clarify whether AKI was defined using established criteria such as KDIGO, and whether persistent reductions in urine output were considered.

Concern #3
It is unclear whether the etiology of AKI was systematically evaluated. Given the polypharmacy common in post-transplant patients, drug-induced nephrotoxicity is one of the plausible cause. If AKI is drug-related, discontinuation of the offending agent alone may suffice for renal recovery. The authors should clarify whether their intervention is intended to be generalized to all AKI cases, or only specific subtypes.

Concern #4
The authors mention that immunosuppressive regimens were modified in some cases within both the non-induction basiliximab (NIB) and matched groups. It would be helpful to present a detailed comparison of the immunosuppressive therapies used in these two groups, ideally in tabular form, and to statistically assess any differences. If significant differences are found, these may confound the interpretation of renal outcomes and should be discussed accordingly.

Author Response

1.The most significant limitation of this study is the small sample size. Although matching was used to select control subjects, the choice of matching parameters is inherently subject to investigator bias, making it difficult to fully eliminate selection bias. From a clinical standpoint, the sample size is too limited to draw definitive conclusions regarding the efficacy of the basiliximab-based regimen on renal function in the setting of AKI. The authors are encouraged to either substantially increase the sample size or justify the adequacy of the current sample through power analysis or similar methodology.

We appreciate these considerations and agree; a limitation of our study is limited sample size and selection bias. We address this in lines 331-332 of our discussion. We included all patients cared for at our transplant center with this NIB intervention, so sample size cannot be further increased.  With that said, the practice of non-induction basiliximab for renal recovery is an innovative approach, and this was intended to be a foundational study to assess the benefits and identify safety concerns.  Propensity matching was explored, but determined unfeasible by our statistician (discussed lines 343-344). 

2.The definition of AKI used in the study appears to be insufficiently rigorous. A transient increase in serum creatinine (e.g., 0.3 mg/dL) can occur with dehydration and may resolve with simple hydration, which would not reflect true renal injury. The authors should clarify whether AKI was defined using established criteria such as KDIGO, and whether persistent reductions in urine output were considered.

A reference to the KDIGO guidelines was added for clarity on lines 87-88. A strength of this study was that it was not dependent on diagnosis codes, but independent assessment of creatinine levels to determine presence and stage of AKI. Urine output was not included for consideration because of the unreliability of this measurement in both hospital and outpatient settings.

3.It is unclear whether the etiology of AKI was systematically evaluated. Given the polypharmacy common in post-transplant patients, drug-induced nephrotoxicity is one of the plausible cause. If AKI is drug-related, discontinuation of the offending agent alone may suffice for renal recovery. The authors should clarify whether their intervention is intended to be generalized to all AKI cases, or only specific subtypes.

The approach taken in the presented cases was for any AKI. The standard approach would be to withdraw offending agents, of which tacrolimus is a known nephrotoxic agent. We systematically assessed for potential physiologic causes of AKI and other nephrotoxins, described in lines 154-161. This study did not seek to assess the causes of AKI in thoracic transplant recipients on tacrolimus, but rather to describe a novel management strategy that could be applied to transplant recipients experiencing AKI. 

4.The authors mention that immunosuppressive regimens were modified in some cases within both the non-induction basiliximab (NIB) and matched groups. It would be helpful to present a detailed comparison of the immunosuppressive therapies used in these two groups, ideally in tabular form, and to statistically assess any differences. If significant differences are found, these may confound the interpretation of renal outcomes and should be discussed accordingly.

This information has been added to the end of Table 2. With a significant comparison noted, a statement has been added to the discussion on lines 293-298. 

Round 2

Reviewer 3 Report

Comments and Suggestions for Authors

The manuscript has been revised. Reasonable explanations were provided for the issues that could not be fully addressed.  There are no more comments.

Author Response

Comment 1: The manuscript has been revised. Reasonable explanations were provided for the issues that could not be fully addressed.  There are no more comments.

Response 1: We thank the reviewer for providing insightful feedback to enhance our manuscript and value to the journals readership.